# Graphene Quantum Dots from Carbonized Coffee Bean Wastes for Biomedical Applications

**DOI:** 10.3390/nano11061423

**Published:** 2021-05-28

**Authors:** Dong Jin Kim, Je Min Yoo, Yeonjoon Suh, Donghoon Kim, Insung Kang, Joonhee Moon, Mina Park, Juhee Kim, Kyung-Sun Kang, Byung Hee Hong

**Affiliations:** 1Program in Nano Science and Technology, Graduate school of Convergence Science and Technology, Seoul National University, Seoul 08826, Korea; dj.kim@snu.ac.kr; 2Graphene Research Center, Advanced Institute of Convergence Technology, Suwon 16229, Korea; yjsuh123@seas.upenn.edu; 3Department of Chemistry, Seoul National University, Seoul 08826, Korea; jyoo3487@bio-graphene.com (J.M.Y.); mn110992@snu.ac.kr (M.P.); juliejjmom@snu.ac.kr (J.K.); 4Graphene Square Inc., Suwon 16229, Korea; 5BIOGRAPHENE, Los Angeles, CA 90013, USA; dkim@dau.ac.kr; 6Department of Pharmacology, Peripheral Neuropathy Research Center (PNRC), Dong-A University College of Medicine, Busan 49201, Korea; 7Adult Stem Cell Research Center and Research Institute for Veterinary Science, College of Veterinary Medicine, Seoul National University, Seoul 08826, Korea; 95teza@naver.com (I.K.); kangpub@snu.ac.kr (K.-S.K.); 8Division of Analytical Science Research, Korea Basic Science Institute (KBSI), Daejeon 34133, Korea; junnymoon@kbsi.re.kr

**Keywords:** graphene, graphene quantum dots, nanomedicine, Parkinson’s disease

## Abstract

Recent studies concerning graphene quantum dots (GQDs) focus extensively on their application in biomedicine, exploiting their modifiable optical properties and ability to complex with various molecules via π–π or covalent interactions. Among these nascent findings, the potential therapeutic efficacy of GQDs was reported against Parkinson’s disease, which has to date remained incurable. Herein, we present an environmentally friendly approach for synthesizing GQDs through a waste-to-treasure method, specifically from coffee waste to nanodrug. Consistent with the previous findings with carbon fiber-derived GQDs, the inhibitory effects of coffee bean-derived GQDs demonstrated similar effectiveness against abnormal α-synuclein fibrillation and the protection of neurons from relevant subcellular damages. The fact that a GQDs-based nanodrug can be prepared from a non-reusable yet edible source illustrates a potential approach to convert such waste materials into novel therapeutic agents with minimal psychological rejection by patients.

## 1. Introduction

Coffee is one of the most widely consumed commodities with a global production of 151.3 million bags annually [1]. Even after dissolving soluble components through extraction, nearly half of the beans remain as waste [1,2]; the production/consumption of coffee instantly raises a subsequent problem of waste disposal. As accumulated coffee grounds are known to be associated with diverse environmental issues, provoked by different substances including caffeine, phenols, tannins, and organic acids [3,4], handling them with an appropriate disposal method is essential. To that end, various approaches have been devised to utilize the waste, including the production of biomass, bioethanol, and fuel in industrial boilers [1,4,5].

In this communication, we expand on the potential areas of coffee waste usage by reporting the preparation of carbonized coffee bean-derived graphene quantum dots (C-GQDs) as a nanodrug candidate for neurodegenerative diseases. In 2018, our group pioneered an application of carbon fiber-derived GQDs as a new treatment option for Parkinson’s disease [6]. Initially, researchers focused on the fluorescence of GQDs and utilized the optical characteristics for relevant applications. However, their structural property has shown an ability to inhibit abnormal α-synuclein (α-syn) fibrillation and disaggregate mature α-syn fibrils, the pivotal process manifesting in the pathogenesis of Parkinson’s disease. In addition, GQDs permeate the blood-brain barrier (BBB) without significant toxicity, a feat that other previous neuro-drug candidates have failed to achieve. When viewed collectively, these findings demonstrate the tremendous potential of utilizing graphene-based nanomaterials as a novel treatment pathway for neurodegenerative diseases. Conventionally, GQDs have been readily synthesized from different carbon precursors, including graphene oxides (GOs) and carbon fibers [7,8,9,10,11,12,13]. Nevertheless, we sought to exploit carbonized coffee waste as a natural, edible source to generate a nanodrug for neurodegenerative diseases, and minimize the potential psychological rejection by patients, all while exploiting the environmentally friendly features of this process [14].

## 2. Materials and Methods

### 2.1. Synthesis of C-GQDs 

Kenya AA, Ethiopia, Mandheling coffee bean compounds were washed with DI water 3 times and thoroughly dried. The compounds were then subjected to carbonization in a chemical vapor deposition (CVD) chamber at 1000 °C for 1 h. The carbonized samples were placed in separate round-bottom flasks to induce with thermo-oxidative cutting in a mixture of H_2_SO_4_ and HNO_3_ (3:1) at 100 °C. Following the reaction, the solution was diluted with DI water and was dialyzed in a 1 kD nitrocellulose membrane tube (Fisher Scientific, Waltham, MA, USA) to extract miniscule debris. Larger fragments were extracted through an inorganic membrane filter (Whatman-Anodisc 47, GE Healthcare, Wauwatosa, WI, USA). Lyophilization of the purified solution resulted in the formation of C-GQD in powder form.

### 2.2. X-ray Photoelectron Spectroscopy (XPS)

XPS was performed using an AXIS-His instrument to investigate the chemical composition of the C-GQDs samples.

### 2.3. Transmission Electron Microscopy (TEM) 

A monolayer graphene film was placed on a 300-mesh lacey carbon-coated copper grid (Ted Pella, Inc., Redding, CA, USA). Subsequently, single droplets of GQDs and C-GQDs (10 μg/mL) were blotted and left to air dry. Each sample was imaged on the HR-TEM (JEM-3010, JEOL, Ltd., Tokyo, Japan). Images were retrieved on the Gatan Digital Camera (MSC-794, Gatan, Inc., Pleasanton, CA, USA).

### 2.4. Raman Spectroscopy 

The Raman spectra were obtained using the Raman spectrometer (Renishaw, Gloucestershire, UK). The C-GQD powder samples on SiO_2_ were exposed to Ar laser (514 nm) for 10 s.

### 2.5. Fourier Transform-Infrared Spectroscopy (FT-IR)

After completely drying the C-GQD samples, the FT-IR spectra were recorded (Nicolet 6700, Thermo Scientific, San Jose, CA, USA) by the conventional KBr pellet technique.

### 2.6. Isolation and Culture of hUCB-MSCs 

With maternal consent, umbilical cord blood (UCB) samples were obtained from the umbilical vein immediately after delivery. All procedures were approved by the Boramae Hospital Institutional Review Board (IRB) and the Seoul National University IRB (IRB no. E1507/001-011). The human umbilical cord blood-derived mesenchymal stem cells(hUCB-MSCs) were isolated and cultured, as previously described. Briefly, the UCB samples were mixed with HetaSep solution (StemCell Technologies, Vancouver, Canada) at a ratio of 5:1 and incubated at room temperature to deplete the erythrocytes. The supernatant was carefully collected, and the mononuclear cells were obtained using Ficoll density-gradient centrifugation at 2500 rpm for 20 min. The cells were washed twice in phosphate-buffered saline (PBS). Cells were seeded at a density of 2 × 10^5^ to 2 × 10^6^ cells/cm^2^ on plates and grown in growth medium made from a KSB-3 Complete Medium kit (Kangstem Biotech, Seoul, Korea) and 10% fetal bovine serum (Gibco BRL, MA, USA). After 3 days, adherent cells formed colonies, and non-adherent cells were removed. For long-term cultures, the cells were seeded at a density of 4 × 10^5^ cells/10-cm plate, and the cells were subcultured upon reaching 80~90% confluency. For hypoxic cultures, cells were cultured in an incubator maintained at 5% CO_2_ and 1% O_2_ for at least 3 days using the delivery of nitrogen gas (N_2_) from a tank containing pure N_2_.

### 2.7. Generation of iNSCs through Direct Conversion 

Induced neural stem cells (iNSCs) were generated from normal donor skin fibroblasts (GM05659; Coriell Institute for Medical Research, Camden, NJ, USA) and NPC patient-derived human Dermal Fibroblasts (hDFs, GM03123 (NPC1P237S/I1061T), GM18453 (NPC1I1061T/I1061T); Coriell Institute for Medical Research). Viral production and transduction were performed as described previously. Briefly, retroviral pMX-SOX2 and pMX-HMGA2 were transfected into 293 FT cells along with VSV-G and gag/pol plasmids using Fugene 6 transfection reagent (Roche, Indianapolis, IN, USA). The viral supernatants were collected at 48- and 72-h post-transfection and used to infect hDFs with 5 μg/mL polybrene (Sigma-Aldrich, St. Louis, MO, USA). For neural stem cell induction, the medium was changed to NSC maintenance medium (ReNcell NSC maintenance media; Millipore, Billerica, MA, USA) with basic fibroblast growth factor (bFGF; Sigma-Aldrich, St. Louis, MO, USA) and epidermal growth factor (EGF; Sigma-Aldrich, St. Louis, MO, USA) after expansion of the infected cells. NSC-like colonies were picked and cultured in neurosphere culture conditions. To generate a homogenous population of iNSCs, cells were maintained as neurospheres and cultured as attached cells on PLO/FN-coated dishes, repeatedly. NPC-iNSC lines from NPC1 mutant human fibroblast were generated up to 10 independent clones.

### 2.8. Cell Viability Assay

The proliferative potential of the hUCB-MSCs was evaluated based on the ability of live cells to convert a tetrazolium salt into purple formazan using the 3-(4,5-dimethylthiazol-2-yl)-2,5-diphenyltetrazolium bromide (MTT, Sigma-Aldrich, St. Louis, MO, USA) assay. hUCB-MSCs, HUVECs, iNSCs (20,000 per well) were seeded on 24-well plates. After 24 h of incubation with or without GQDs, 50 mL MTT stock solution (5 mg/mL; Sigma-Aldrich, St. Louis, MO, USA) was added to each well, and the plates were further incubated for 4 h at 37 °C. The supernatant was removed, and 500 μL of DMSO (dimethyl sulfoxide) was added to each well to solubilize the purple formazan crystals. The solution was then transferred to a 96-well microplate for measurement. The absorbance at a wavelength of 540 nm was measured using an EL800 microplate reader (BIO-TEK Instruments, Winooski, VT, USA). All of the measurements were performed in triplicate. AlamarBlue cell viability assay kit was also utilized to assess additional neuron viability (Cat#: DAL1025; Molecular Probes™), following the manufacturer’s instructions.

## 3. Results and Discussion

### 3.1. Synthesis of C-GQDs

Figure 1 shows the schematic illustration of the synthesis process of C-GQDs from Ethiopia, Mandheling, and Kenya AA coffee beans. After dissolving and discarding soluble impurities with deionized water, the coffee beans were carbonized through hydrogen annealing at 1000 °C (Figure 1a). Black, coal-like carbonaceous precursors were obtained after the carbonization process, as opposed to the coffee beans’ original brown color (Figure 1f). In like manner with the carbon fiber-derived methods [10], C-GQDs were directly synthesized from carbonized coffee beans via thermo-oxidative cutting in a mixture of strong acids (H_2_SO_4_:HNO_3_ = 3:1) at 100 °C for 24 h, with subsequent purification steps (Figure 1b–d). Figure 1e illustrates the representative structure of C-GQDs with the graphitic plane composed of pyrrolic, pyridinic, and graphitic nitrogen that are originated from the intrinsic nitrogen-containing functional groups of coffee beans.

To examine the size distributions of C-GQDs to confirm their consistency against conventional GQDs, TEM and atomic force microscopy (AFM) analyses were carried out. Figure 2a–d shows the TEM images manifesting the average lateral dimensions of carbon fiber GQDs (CF-GQDs), Ethiopia GQDs (E-GQDs), Mandheling GQDs (M-GQDs), and Kenya AA GQDs (K-GQDs) to be 2.60 ± 0.58 nm, 3.02 ± 0.913 nm, 3.66 ± 0.84 nm, and 3.69 ± 0.98 nm, respectively. On average, the particle size of C-GQDs is 0.4~1 nm larger than CF-GQDs. The crystallinity of the basal planes of these C-GQDs was validated by the presence of lattice fringes in the selected area electron diffraction pattern (SAED) analyses (Figure 2e–h). It should be noted that the lattice spacings were 0.24 nm, 0.21 nm, and 0.31 nm respectively for E-GQDs, M-GQDs, and K-GQDs, which correspond to that of CF-GQDs (0.31 nm) and the reported values for graphitic materials [15,16]. The average heights were measured by AFM, showing 3.04 ± 0.32 nm, 3.23 ± 0.41 nm, and 3.18 ± 0.38 nm respectively for E-GQDs, M-GQDs, and K-GQDs (Appendix A).

The optical properties of C-GQDs were investigated by photoluminescence (PL), Raman, Fourier-transform infrared (FT-IR) spectroscopy analyses. PL analysis allows further comparison of C-GQDs’ fluorescence originated from subtle differences in size and functional groups. The PL spectra of C-GQDs show two strong peaks corresponding to the π–π* and σ–π* transitions, as described in the literature (Appendix A) [17,18]. Interestingly, the photoluminescence of C-GQDs varies by their origins, showing the maximum emissions at 540 nm, 490 nm, and 530 nm, respectively for E-GQDs, M-GQDs, and K-GQDs, which can also be visually reflected by ambient exposure to a 365 nm UV lamp in deionized water (Figure 3a). Such discrepancy can be attributed to the difference in their atomic compositions, which consequentially results in heteroatom doping effects in the synthesized C-GQDs (Appendix A) [19,20]. Raman spectroscopic analysis verifies the presence of graphitic domain in the synthesized C-GQDs. The Raman spectra of all types of C-GQDs exhibit the G and D peaks respectively at 1590 cm^−1^ and 1370 cm^−1^, which are the characteristic bands of graphene-based materials (Figure 3b,c) [21,22]. Noticeably, the same peaks are present yet barely observable in carbonized precursors (Appendix A). This can be attributed to the presence of sp^3^ carbons in the precursors, which undergo decomposition during the synthesis process; energetically more stable sp^2^ graphitic domains remain intact and thus result in more intense Raman signals [23].

The FT-IR spectra-based functional group analysis displays distinguished peaks for hydroxyl, amide, carboxyl groups, and aromatic carbon double bonds (Figure 3c) [24]. Among these peaks, the relative intensities of carboxyl groups (–COOH) and aromatic carbon double bonds (C=C), respectively at 1720 cm^−1^ and 1620 cm^−1^ are deemed as an important barometer for GQDs to fully elicit therapeutic efficacy against Parkinson’s disease. Not only do carboxyl groups play a vital role in the interaction between GQDs and α-syn, but their relative abundance is also decisive to minimize toxicity both in vitro and in vivo. We thus strongly believe that the ratio between C=O and C=C over/around 1:1 is desirable for therapeutic applications; the values for C-GQDs are comparable to that of carbon fiber-derived GQDs, which show exceptional therapeutic effects without significant toxicity [6]. The presence of carboxyl groups on C-GQDs was further validated by X-ray photoelectron (XPS) analysis, which indicates strong C=O and C–O signals at 288.5 eV and 286.5 eV, respectively.

### 3.2. Therapeutic Efficacy of C-GQDs for Parkinson’s Disease 

The therapeutic efficacy of C-GQDs for Parkinson’s disease was investigated in primary neurons (Figure 4). The pathogenesis of Parkinson’s disease is strongly associated with abnormal fibrillation of α-syn in the midbrain and the fibril aggregates spreading throughout the entire region to provoke loss of dopamine-producing neurons [25,26,27,28]. As an exemplary sporadic in vitro model, the treatment of α-syn preformed fibrils (α-syn PFFs) leads to accelerated α-syn fibrillation and subsequent cell death in neurons. The presence of C-GQDs, however, significantly inhibited α-syn fibrillation and correlated neuron toxicity and, as assessed by AlamarBlue assay and phosphorylated α-syn (p-α-syn) immunohistochemical analysis. It should be noted that we could not observe any critical differences in the efficacy of CF-GQDs and C-GQDs. The results imply that the presence of nitrogen atoms in C-GQDs does not interfere with GQDs’ therapeutic function as long as their size and functional groups remain similar.

## 4. Conclusions

In this report, we demonstrated the synthesis of C-GQDs from carbonized coffee bean waste, via a waste-to-treasure strategy. The TEM and AFM analyses showed that C-GQDs can be successfully obtained in a size-controllable manner through carbonization and thermo-oxidative cutting processes. In addition, the optical properties of as-prepared C-GQDs were further investigated by PL, Raman, FT-IR, and XPS analyses, which could also show the difference in heteroatom contents varied by the origins of the coffee beans. Furthermore, C-GQDs also exhibit negligible toxicity in primary neurons and the therapeutic efficacy for Parkinson’s disease was practically the same as that of CF-GQDs. As naturally derived drugs may prove more conducive to patient compliance, our results suggest an environmentally friendly and efficient method for a new synthesis of nanomedicine.

## Figures and Tables

**Figure 1 nanomaterials-11-01423-f001:**
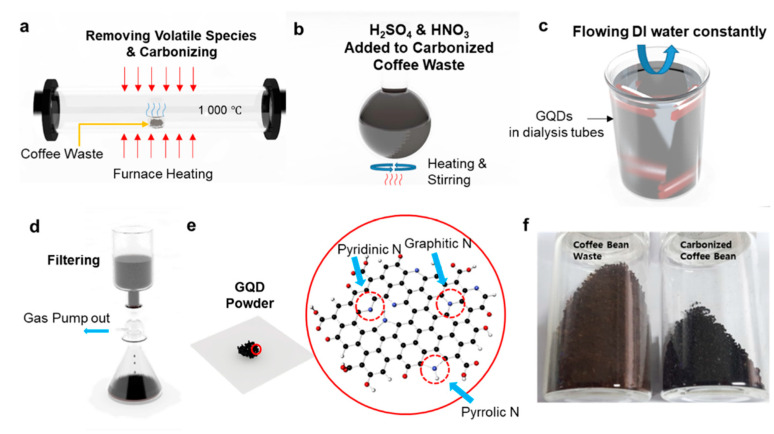
Schematic illustration of the synthesis of coffee bean-derived GQDs, (**a**) Carbonization process to remove undesirable chemicals in coffee beans at 1000 °C with hydrogen gas, (**b**) The resulting carbon precursor was refluxed in H_2_SO_4_ and HNO_3_ (3:1) solution at 100 °C, (**c**) The obtained solution is dialyzed in a dialysis bag to remove acid, (**d**) To discard unreacted carbon debris, the solution was vacuum filtered with a membrane filter, (**e**) Schematic illustration of as-synthesized coffee bean-derived GQDs. The graphitic plane of the GQDs consists of pyrrolic, pyridinic, and graphitic nitrogen, respectively, (**f**) Photo images of coffee beans pre- and post-carbonization process.

**Figure 2 nanomaterials-11-01423-f002:**
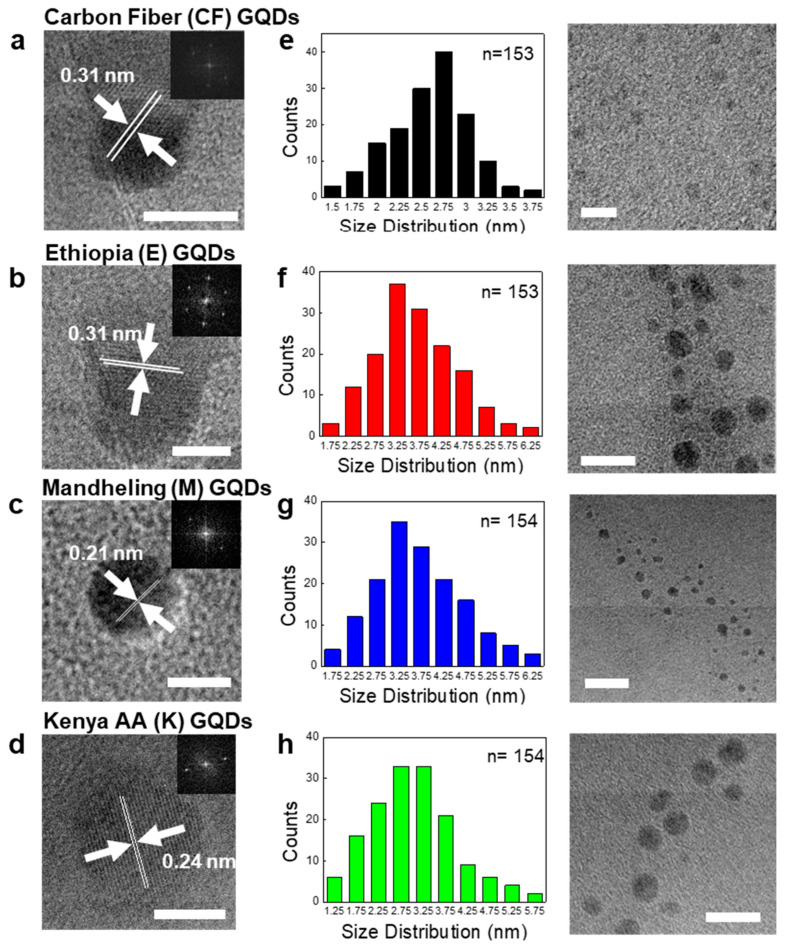
TEM images, size distributions of the coffee bean-derived GQDs. (**a**–**d**) Representative HR-TEM images of the carbon fiber-derived GQDs and the coffee bean GQDs with inter-layer spacing (scale bar = 5 nm). The SAED patterns are shown in the inset. (**e**–**h**) Representative size distributions of carbon fiber-derived GQDs and the coffee bean GQDs.

**Figure 3 nanomaterials-11-01423-f003:**
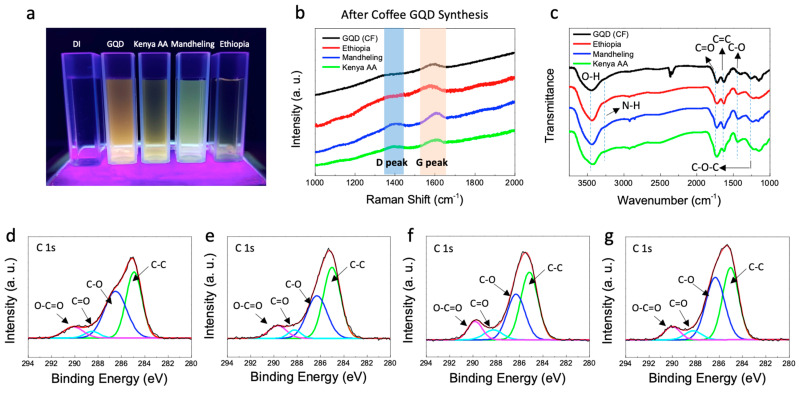
Optical characterization of coffee bean-derived GQDs. (**a**) Representative image of the photoluminescence of GQDs solutions under UV (365 nm). (**b**,**c**) Raman spectra and FT-IR of the carbon fiber-derived GQDs and the coffee bean-derived GQDs. (**d**–**g**) C 1s XPS spectra of the carbon fiber-derived GQDs and the coffee bean-derived GQDs.

**Figure 4 nanomaterials-11-01423-f004:**
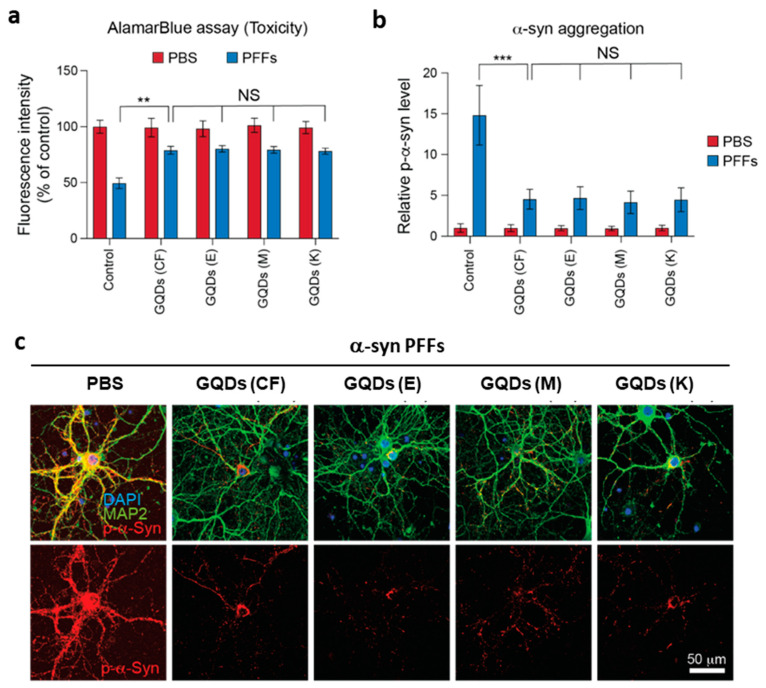
The effect of coffee bean-derived GQDs on α -syn PFFs-induced primary neuronal toxicity and α-syn pathology. (**a**) The mouse cortical neurons (10 DIV) were treated with 5 μg/mL of α-syn PFFs with carbon fiber derived GQDs (CF, 5 μg/mL, *n =* 6) or coffee bean-derived GQDs (E: Ethiopia, 5 μg/mL, *n =* 6; M: Mandheling, 5 μg/mL, *n =* 6; K: Kenya AA, 5 μg/mL, *n =* 6). Neuronal toxicities were measured by alamarBlue assay at 7 days after treatment. (**b**,**c**) Representative p-α-syn immunostained neurons at 7 days after treatment with p-α-syn antibody. The p-α-syn immunofluorescence intensities were assessed and normalized to the PBS control (*n =* 6, each group). Significant levels are denoted as asterisks: * *p* < 0.05, ** *p* < 0.01, and *** *p* < 0.001 via one-way ANOVA with Tukey’s multiple comparisons test.

## Data Availability

The data presented in this study are available on request from the corresponding author.

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
