# Peer review of "Graphene Quantum Dots from Carbonized Coffee Bean Wastes for Biomedical Applications"

_nanomaterials, 2021, doi:10.3390/nano11061423_

Round 1
Reviewer 1 Report
The communication manuscript by Kim et al discusses a green manufacturing procedure of C-GQDs from carbonised coffee waste and highlighted its potential Parkinson disease. This work is very interesting and has great potential in nanomedicine and merit publication in Nanomaterials as it is.
Author Response
Thank you very much for the positive evaluation on our manuscript. We are genuinely excited about the opportunity to share our findings through Nanomaterials.
Reviewer 2 Report
In this manuscript, the author reports, ‘Graphene quantum dots from carbonized coffee bean wastes for biomedical applications’. Herein the author demonstrated coffee bean derived graphene quantum dots as a nanodrug candidate for neurodegenerative diseases. The authors should address the following minor questions before getting a possible publication.
- In the ‘2.1. Synthesis of C-GQDs’ section, the author should mention the pore size of the membrane filter. What is the time period for dialysis?
- The author should write purpose for each test in one/two sentences (in brief) before explaining the results of the characterization techniques. Therefore, the logic and organization of this part will be enhanced.
- The authors are encouraged to perform UV-vis absorption spectra of the GQDs.
- The formatting and grammatical errors in the article need to be checked carefully by professional proof readers.
- What does the error bars stand for in Fig.4? It should be mentioned in the figure caption.
- Abstract should be compact, precise and should present brief information about the work.
- Why the absorbance value for Kenya in Figure S6c is low in comparison to others?
- The authors cited some of the relevant research works that have been conducted in this area however there are a few that needs to be included for better literature review: Research on Chemical Intermediates 45.7 (2019): 3823-3853; Particle & Particle Systems Characterization 31.4 (2014): 415-428; Nanomaterials 11.2 (2021): 369.

Author Response
Thank you very much for your valuable feedback and comments. Below are the point-by-point responses for the raised issues.
- In the ‘2.1. Synthesis of C-GQDs’ section, the author should mention the pore size of the membrane filter. What is the time period for dialysis?
- The pore size is stated in the section – ‘1 kD’ nitrocellulose membrane tube. The dialysis process was carried out for 3 days with continuous water flow.
- The author should write purpose for each test in one/two sentences (in brief) before explaining the results of the characterization techniques. Therefore, the logic and organization of this part will be enhanced.
- Thank you for your suggestion. The purpose of each test in the relevant lacking areas has been supplemented in the revised manuscript.
- The authors are encouraged to perform UV-vis absorption spectra of the GQDs.
- Thank you for your feedback. Currently, the only available sample is Kenya-GQD, and it is difficult to produce the GQDs samples before the revision deadline, as the process takes several days. However, we want to emphasize that the optical properties are not decisive in eliciting the therapeutic efficacy against Parkinson’s disease. In addition, we believe other results (including UV lamp pics) pertaining to the optical properties provide insight into the general tendency of the samples. For your review, the UV-vis absorption spectrum of Kenya-GQD is attached below:
- The formatting and grammatical errors in the article need to be checked carefully by professional proof readers.
- We have carefully reviewed the manuscript further with a native speaker/professional writer as stated in the acknowledgements section.
- What does the error bars stand for in Fig.4? It should be mentioned in the figure caption.
- Supplementary information about the significance levels have been added in the revised manuscript.
- Abstract should be compact, precise and should present brief information about the work.
- Thank you for your suggestion and we have carefully reviewed and revised the abstract.
- Why the absorbance value for Kenya in Figure S6c is low in comparison to others?
- Compared to other cell lines like hUCB-MSC or HUVEC, the induced neural stem cell (INSC) was more sensitive to GQD treatment. Currently, it is difficult to elucidate why Kenya AA showed lower viability in INSC compared to hUCB-MSC or HUVEC. However, this may be attributed to a unique by-product of Kenya AA generated during GQD synthesis. Although the coffee beans go through the same manufacturing process, each coffee possess distinct structural property. The structural uniqueness of Kanya AA GQDs could trigger a specific signaling pathway and influence cell proliferation through an unknown mechanism. Thus, it would be interesting to further investigate the role of detailed structures and the correlation with the biology of neuronal stem cell.
- The authors cited some of the relevant research works that have been conducted in this area however there are a few that needs to be included for better literature review: Research on Chemical Intermediates 45.7 (2019): 3823-3853; Particle & Particle Systems Characterization 31.4 (2014): 415-428; Nanomaterials 11.2 (2021): 369
- The suggested references have been added in the revised manuscript:
Conventionally, GQDs have been readily synthesized from different carbon precursors including graphene oxides (GOs) and carbon fibers [7-13].
Reference numbers were also changed, and the changes are marked in blue in the main text.
